# Late quaternary biotic homogenization of North American mammalian faunas

Danielle Fraser [1,2,3,4✉], Amelia Villaseñor[5], Anikó B. Tóth [6], Meghan A. Balk[7], Jussi T. Eronen[8,9], W. Andrew Barr [10], A. K. Behrensmeyer[4], Matt Davis[11], Andrew Du[12], J. Tyler Faith[13,14], Gary R. Graves [15,16], Nicholas J. Gotelli[17], Advait M. Jukar[4,18,19], Cindy V. Looy [20], Brian J. McGill[21], Joshua H. Miller [22], Silvia Pineda-Munoz[23], Richard Potts [24], Alex B. Shupinski[25], Laura C. Soul[4] & S. Kathleen Lyons [25]

Biotic homogenization—increasing similarity of species composition among ecological communities—has been linked to anthropogenic processes operating over the last century. Fossil evidence, however, suggests that humans have had impacts on ecosystems for millennia. We quantify biotic homogenization of North American mammalian assemblages during the late Pleistocene through Holocene (~30,000 ybp to recent), a timespan encompassing increased evidence of humans on the landscape (~20,000–14,000 ybp). From ~10,000 ybp to recent, assemblages became significantly more homogenous (>100% increase in Jaccard similarity), a pattern that cannot be explained by changes in fossil record sampling. Homogenization was most pronounced among mammals larger than 1 kg and occurred in two phases. The first followed the megafaunal extinction at ~10,000 ybp. The second, more rapid phase began during human population growth and early agricultural intensification (~2,000–1,000 ybp). We show that North American ecosystems were homogenizing for millennia, extending human impacts back ~10,000 years.

[1] Palaeobiology, Canadian Museum of Nature, Ottawa, ON, Canada. [2] Biology, Carleton University, Ottawa, ON, Canada. [3] Earth Sciences, Carleton University, Ottawa, ON, Canada. [4] Department of Paleobiology and Evolution of Terrestrial Ecosystems Program, Smithsonian Institution, National Museum of Natural History, Washington, DC, USA. [5] Department of Anthropology, University of Arkansas, Fayetteville, AR, USA. [6] Centre for Ecosystem Science, School of Biological, Earth and Environmental Sciences, University of New South Wales, Sydney, NSW, Australia. [7] National Ecological Obervatory Network, Battelle Memorial Institute, Boulder, CO, USA. [8] Ecosystems and Environment Research Programme & Helsinki Institute of Sustainability Science, Faculty of Biological and Environmental Sciences, University of Helsinki, Helsinki, Finland. [9] BIOS Research Unit, Helsinki, Finland. [10] Center for the Advanced Study of Human Paleobiology, Department of Anthropology, The George Washington University, Washington, DC, USA. [11] Natural History Museum of Los Angeles County, Los Angeles, CA, USA. [12] Department of Anthropology and Geography, Colorado State University, 1787 Campus Delivery, Fort Collins, CO, USA. [13] Natural History Museum of Utah, University of Utah, Salt Lake City, UT, USA. [14] Department of Anthropology, University of Utah, 260S, Central Campus Drive, Salt Lake City, UT, USA. [15] Department of Vertebrate Zoology, National Museum of Natural History, Smithsonian Institution, Washington, DC, USA. [16] Center for Macroecology, Evolution and Climate, Globe Institute, University of Copenhagen, Copenhagen, Denmark. [17] Department of Biology, University of Vermont, Burlington, Vermont, USA. [18] Yale Institute for Biospheric Studies, Yale University, New Haven, CT, USA. [19] Department of Anthropology, Yale University, New Haven, CT, USA. [20] Department of Integrative Biology and Museum of Paleontology, University of California, Berkeley, Valley Life Sciences Building, Berkeley, CA, USA. [21] School of Biology and Ecology and Mitchell Center for Sustainability Solutions, University of Maine, Orono, ME, USA. [22] Department of Geology, University of Cincinnati, Cincinnati, OH, USA. [23] Department of Earth and Atmospheric Sciences, Indiana University, Bloomington, IN, USA. [24] Human Origins Program, National Museum of Natural History, Smithsonian Institution, Washington, DC, USA. [25] School of Biological Sciences, University of Nebraska Lincoln, Lincoln, NE, USA. ✉email: dfraser@nature.ca

The global-scale ecological impacts of humans have accelerated over the last several thousand years[1–3]. Human-driven climate changes and exploitation of natural resources, as well as increasing globalization and urbanization linked to rapid population growth, are now altering landscapes to such a degree that few pristine ecosystems remain[1,4,5]. Consequences of these anthropogenic stressors include widespread extirpation of species[6,7], population declines[4], geographic range shifts and expansions[8], significant ecological downgrading[9], and an impending wave of global extinctions[10]. At the local scale, however, meta-analyses reveal no net decreases in species richness[11–13], while regional studies document biotic homogenization (i.e., increased similarity in the composition of species among ecological communities, also referred to as reduced β diversity) for several extant groups including birds, fishes[14], mammals[7,15–17], and plants[13,18,19], in terrestrial and aquatic settings over the past one hundred years[13,15,17,20–22].

Biotic homogenization can result from increased species coexistenc[21], driven by factors including, but not limited to, range expansions, human-mediated species translocation, and agricultural landscape modification (linked to decreased landscape heterogeneity, increased patch size, and the intentional or unintentional spread of generalist, competitively dominant native or non-native species)[6,8,13,19,20,23]. It can also result from the extinction or extirpation of endemics[7,24,25], which similarly reduces the uniqueness of spatially separated species assemblages[15,21] (Supplementary Fig. 1). Ongoing biotic homogenization is a conservation concern because range-expanding and introduced species, as well as those that are displaced or at risk of extinction, may be clustered in particular functional groups (i.e., groupings based on the roles they play in ecosystems)[21,26,27]. For example, extant mammal species at risk of extinction are commonly large-bodied carnivores[28]. Such non-random biodiversity loss could lead to profound modifications in the nature of biotic interactions (e.g., trophic interactions), altering the ways in which materials and energy flow through ecosystems[26,29], reducing their resilience to ongoing and future perturbation[9]. In particular, the replacement of keystone species—those whose impacts on interactions and the environment are outsized with respect to their relative abundance—by species whose impacts are proportional to their relative abundance, has broad implications for associated biota (e.g., through the loss of certain ecosystem engineering processes)[30] that can include reduction in ecosystem functioning and production of ecosystem services[31,32].

Until now, however, biotic homogenization has been studied almost exclusively on relatively short time scales that encompass the last few decades to the past century (e.g.,[7,13,15,19]). Although historical biodiversity data are valuable for understanding the effects of accelerating anthropogenic climate and landscape change (e.g.,[13]), these short-term data provide only a partial, time-limited picture of biodiversity change and may incorrectly identify drivers[33–35]. In particular, they do not provide ecological scenarios in which human influence is minor or negligible (often termed baselines) against which observed changes can be compared[36], making it difficult to fully understand the magnitude and timing of large-scale human effects on biodiversity. Understanding the deeper-time history of biotic homogenization is therefore critical to better define the contributions of anthropogenic and non-anthropogenic processes.

Evidence is mounting that humans have had landscape-scale effects on ecosystems for thousands of years[6,37,38]. The dispersal of humans into North America (~16,000 –14,000 ybp and possibly prior to 20,000 ybp)[39,40], while coincident with significant climatic perturbation[39,41], was also associated with the loss of 72% of large-bodied (>44 kg) mammal genera[42–45]. Much later,

humans undertook the development of geographically extensive agriculture in North America (starting 2,000–1,000 ybp)[37]. The net result has been significant ecological perturbation, leading to shifts in species distributions[46], major changes in mammal community structure[47–53], and changes to fundamental biotic interactions[54–57]. To our knowledge, biotic homogenization has yet to be studied for terrestrial North American mammals at the continental scale and over timescales incorporating the late Pleistocene and entire Holocene (~30,000 ybp to present), a critical period in North American pre-history that encompasses times of both little human influence and strongly anthropogenic scenarios.

We test for biotic homogenization of North American mammalian assemblages during the late Pleistocene and Holocene (~30 ka to modern), including records from before and after the arrival of humans in the Americas (see Methods). Here, we define an assemblage as the co-occurrence of species as recorded by specimens at paleontological or archeological sites, and modern observations records. Our data are taxonomically well-resolved with 8,831 occurrences of 365 species at 366 localities. The data include all extant North American mammalian orders spanning the last 30,000 years divided among six time bins, i.e., 5,000 year intervals between 30,000 ybp and 500 ybp, with the seventh (modern) interval being the 1970–2000s. To minimize bias due to differences in sampling between the modern and fossil records, all included sites were required to have 20 or more species and at least one occurrence each from Artiodactyla, Carnivora, and Rodentia. Our vetting process ensures sampling across a variety of body sizes and representation from clades comprising the majority of non-volant North American mammal diversity (Supplementary Fig. 2).

Biotic homogenization refers to increasing compositional similarity among ecological communities (i.e., declining β diversity), which is frequently quantified as the average of a specified pairwise similarity metric[58]. We quantify change in the mean species compositional similarity for North American mammalian assemblages using the pairwise Jaccard similarity metric [i.e., $J/(A + B\text{-}J)$, where $A$ and $B$ are the number of species present at Site 1 and Site 2, respectively, and $J$ is the number of species shared by the two sites; the metric, varies from zero and one, with zero indicating 100% compositionally distinct assemblages and one indicating compositionally identical assemblages[59]]. We also compare Jaccard similarity to other less commonly-used metrics (e.g., Forbes Similarity[60]), and provide a series of null model expectations designed to account for the effects of spatiotemporal changes in sampling. The null model shuffles sites among time bins, thus providing a measure of the significance of change in mean taxonomic similarity among time bins (see Methods)[61]. We perform the same analyses on subsets of the data to address potential differences among mammalian size classes (species > 1 kg and > 5kg[62–64]) as well extinct and extant species (excluding extinct megafauna[15,21]).

We perform a series of sensitivity analyses to evaluate the effects of varying intensity of fossil record sampling (i.e., number of sites), topographic complexity, and spatiotemporal changes in the distribution of fossil data (Supplementary Fig. 2)[65]. To do so, we perform our analyses on subsets of the data, i.e., equalizing the number of sites among time bins and re-sampling, excluding areas of high topographic complexity, and excluding high latitude samples (see Methods). Topographically complex regions such as the Rocky Mountains show comparatively low mean taxonomic similarity among sites, a phenomenon that is typically attributed to high environmental turnover[66]. Taxonomic similarity among sites also tends to increase with latitude[66,67]. Thus, changes in the intensity of sampling across topographically variable regions or across latitudes may lead to anomalous changes in mean

taxonomic similarity. Our sensitivity analyses are designed to tease apart sampling effects from ecological signal. Graham, et al.[47] performed a broadly similar set of analyses, which we extend to create a comparison to modern mammal assemblages, while considering heterogeneity in spatiotemporal patterns of sampling and addressing differences between mammals from different body size classes.

Finally, we explore potential drivers of biotic homogenization by comparing to temporal changes in climate heterogeneity (i.e., differences in climate among sites measured within each time bin)[68], species geographic range sizes[53], human presence on the North American landscape (~20,000–14,000 ybp)[39], extinction of the mammalian megafauna (beginning ~15,000 ybp and culminating by ~11,700 ybp)[45], and the development of extensive agriculture (~2,000–1,000 ybp)[37]. We hypothesize that there were two periods of significant biotic homogenization, the first following the extinction of the mammalian megafauna (~12,000 ybp–10,000 ybp), and second the development of widespread agricultural activities (2,000–1,000 ybp). Support for our hypothesis would constitute strong evidence for an ancient origin of anthropogenic biotic homogenization and amplify calls for deep time perspectives in the study of human impacts on ecosystems.

## Results

**Mean taxonomic similarity**. During the late Quaternary (30,000 ybp to modern), the average taxonomic similarity of mammalian assemblages was relatively stable and within null expectations (null model generated by shuffling sites among time bins; see Methods) until the Holocene (Fig. 1; Table 1). Mean assemblage similarity increased by 0.15 (Jaccard similarity) from the

10,000–5,000 ybp time bin through to the modern (Fig. 1; black line), occurring at the fastest rate between the final two time bins (5,000–500 ybp and 500 ybp–modern; Fig. 1). Assemblages composed of mammals larger than 1 kg and 5 kg showed the greatest degree of homogenization. They increased in similarity by 0.25 (Jaccard similarity) from the ~15,000–10,000 ybp time bin onward (Fig. 1; dashed lines), becoming more homogenous than null expectations from the 10,000 ybp–5,000 ybp time bin onward (Table 1). Large mammal (>1 kg) assemblages experienced two periods of rapid homogenization, from the 15,000–10,000 ybp bin to the 10,000–5,000 ybp bin and from the 5,000–0.5 ybp bin to the modern (Fig. 1; dashed lines). The same patterns are evident when aligning the time bins with the onset of deglaciation at the beginning of Heinrich Stadial 1 and the Pleistocene-Holocene

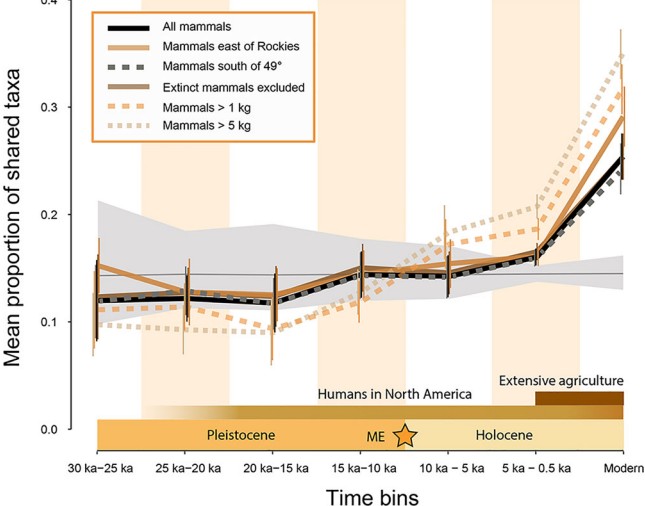

**Fig. 1 Mammal assemblages undergo biotic homogenization during the Holocene (sample sizes in Source Data).** Mammals larger than 1 kg commence homogenizing during the 15,000–10,000 ybp time bin, but the onset of homogenization is delayed until after 10,000 ybp for assemblages including all mammals. Mammals larger than 1 kg are more homogenous than null expectations by the 10,000–5,000 ybp time bin while assemblages of all mammals are more homogenous than null expectations by the 5,000–500 ybp time bin. Change in mean taxonomic similarity (Jaccard similarity index) among sites ± the standard error of the mean. Gray ribbon shows the mean of the null model runs (sites shuffled among time bins) with 95% confidence intervals. Dates of the mammal sites are based on calibrated radiocarbon dates (See Material and Methods). Extinction of the mammal megafauna in North America (ME). The modern time bin (1980's–2010's) is portrayed as larger to enhance readability.

**Table 1 Effect sizes for Jaccard similarity across 30,000 ybp.**

| Metric | Time bin | Effect size |
|---|---|---|
| Jaccard similarity (all species) | 30,000–25,000 | −0.68 |
| | 25,000–20,000 | −1.00 |
| | 20,000–15,000 | −1.11 |
| | 15,000–10,000 | 0.33 |
| | 10,000–5,000 | 0.23 |
| | 5,000–500 | 5.28* |
| | Modern | 14.35* |
| Jaccard similarity (east of Rocky Mountains) | 30,000–25,000 | 0.14 |
| | 25,000–20,000 | −0.74 |
| | 20,000–15,000 | −0.86 |
| | 15,000–10,000 | −0.09 |
| | 10,000–5,000 | 0.61 |
| | 5,000–500 | 3.67* |
| | Modern | 14.85* |
| Jaccard similarity (south of 49th parallel) | 30,000–25,000 | −0.71 |
| | 25,000–20,000 | −0.61 |
| | 20,000–15,000 | −0.99 |
| | 15,000–10,000 | 0.17 |
| | 10,000–5,000 | 0.08 |
| | 5,000–500 | 5.08* |
| | Modern | 10.80* |
| Jaccard similarity (no extinct species) | 30,000–25,000 | −0.75 |
| | 25,000–20,000 | −0.91 |
| | 20,000–15,000 | −0.99 |
| | 15,000–10,000 | 0.45 |
| | 10,000–5,000 | 0.05 |
| | 5,000–500 | 5.40* |
| | Modern | 13.13* |
| Jaccard similarity (species > 1 kg) | 30,000–25,000 | −1.00 |
| | 25,000–20,000 | −1.40 |
| | 20,000–15,000 | −2.34 |
| | 15,000–10,000 | −1.51 |
| | 10,000–5,000 | 2.58* |
| | 5,000–500 | 12.51* |
| | Modern | 21.67* |
| Jaccard similarity (species > 5 kg) | 30,000–25,000 | −1.47 |
| | 25,000–20,000 | −2.49 |
| | 20,000–15,000 | −2.52 |
| | 15,000–10,000 | −0.86 |
| | 10,000–5,000 | 3.46* |
| | 5,000–500 | 18.13* |
| | Modern | 25.66* |

Starred numbers are those where Jaccard similarity fell outside the 95% confidence intervals of the null model.

transition (Supplementary Fig. 3a), suggesting that the pattern is not an artefact of how we grouped sites into time bins.

**Fossil record sampling**. Re-sampling (i.e., randomly drawing the same number of sites for each time bin to homogenize sampling intensity) did not change the overall pattern of Holocene biotic homogenization (Supplementary Fig. 4, S5). Furthermore, mean taxonomic similarity as calculated using Jaccard similarity (1 – Jaccard dissimilarity) is uncorrelated with total within-time-bin species richness (i.e., size of the regional species pool; $p > 0.05$, $R^2 = 4.0 \times 10^{-4}$; Supplementary Fig. 6). Exclusion of sites from the Rocky Mountains westward (to reduce the effects of topographic heterogeneity; see Methods; Fig. 1; lighter brown line) and north of the Canadian border (to address trends in sampling area and density) does not alter our results (Fig. 1; gray dotted line). Thus, changes in spatial and taxonomic sampling are not likely to be responsible for the pattern of Holocene biotic homogenization.

**Additional similarity metrics**. The pattern of Holocene biotic homogenization is apparent for most of the similarity metrics employed herein (Supplementary Fig. 7). The relative stability of nestedness through time suggests that much of the change in mean taxonomic similarity during the Holocene is a result of declining turnover (Supplementary Fig. 7A). The divergent patterns observed for distance decay of similarity and the corrected Forbes Index appear to reflect correlations with the number of sites ($p < 0.05$, $R^2 = 0.79$; Supplementary Fig. 8B) and regional species richness ($p < 0.05$, $R^2 = 0.80$; Supplementary Fig. 8B), respectively.

**Palaeoclimate turnover**. To test for an association between climate heterogeneity and mean mammal taxonomic similarity, we performed a PCA of annual average minimum temperature, annual average maximum temperature, annual actual evapotranspiration (AET), and annual precipitation. We then calculated climate turnover as the mean pairwise climate dissimilarity in PCA scores amongst sites in each time slice. The loadings for both PC1 and PC2 showed that both axes are primarily correlated with annual evapotranspiration (AET) and total annual precipitation (AP) for all time slices. Sampling the climate rasters using the distribution of fossil sites (Fig. 2, solid brown line) shows a more pronounced effect of deglaciation on climate heterogeneity than sampling evenly across the continental USA and southern Canada at 500-year intervals (Fig. 2, solid orange line), simply owing to changes in fossil site distributions among time bins. Both sets of curves, however, show that climate heterogeneity declined between 20,000 ybp and 15,000 ybp with relatively little change thereafter.

**Geographic range size change**. After 10,000 ybp, we observe ~50% increases in mean geographic range size among mammals larger than 1 kg (Fig. 3A) but not increases in range fill (occupancy; Fig. 3B). Assemblages including smaller mammals (<1 kg), however, showed much more moderate increases in geographic range size and biotic homogenization that were delayed until after ~5,000 ybp (Fig. 3A).

## Discussion

While historical biodiversity data are invaluable for understanding recent patterns of biotic homogenization (e.g.,[13,15]), they do not provide pre- or low human impact scenarios and therefore cannot address when humans began to have large-scale ecological impacts[33]. We used the Pleistocene through Holocene (30,000 ybp to present) records of North American mammals to assess

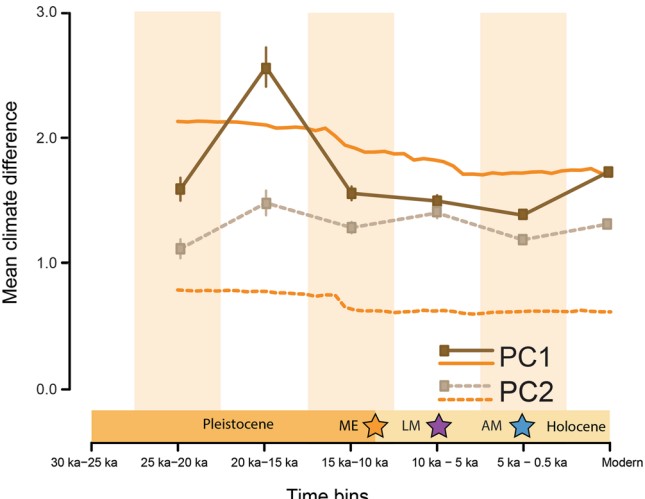

**Fig. 2 Mean climate difference between bins does not decrease during the intervals for which we observe biotic homogenization (sample sizes in Source Data).** Change in climate turnover (mean climate difference) ± the standard error of the mean. Climate estimates are based on de-biased and downscaled earth system model (ESM) climate simulations from recent and paleoclimate models at 0.5 degree resolution[68]. The orange star indicates the extinction of the mammal megafauna in North America (ME). The purple and blue stars indicate points at which assemblages of large mammals (LM) and all mammals (AM) were more homogenous that null expectations. The lines in brown tone indicate climate rasters sampled at the same locations as the fossil and archeological sites included in the study. The lines in orange tone represent sampling of the climate rasters are equal intervals across the landscape between 35 and 80˚N based on the fact that the majority of the fossil and archeological sites included in the present study occur between these latitudes.

the onset of biotic homogenization in North America (Supplementary Fig. 2) and address the potential roles of human dispersal (~20,000–14,000 ybp), extinction of the mammal megafauna (~12,000–10,000 ybp), and acceleration of human impacts (e.g., development of extensive agriculture; ~2,000–1,000 ybp). The present study provides, to the best of our knowledge, the most temporally and taxonomically inclusive as well as spatially extensive study of mammalian biotic homogenization to date. We observe mammal assemblages that are homogenized, i.e., more similar than null expectations, as early as 10,000–5,000 ybp, with biotic homogenization commencing between 15,000 and 10,000 ybp for mammals larger than 1 kg and 10,000–5,000 ybp for all mammals (Fig. 1; Table 1). Our various sensitivity and re-sampling scenarios do not change the overall pattern (Fig. 1; Supplementary Figs. 3–6), indicating that this is not an artefact of spatiotemporal changes in fossil record sampling, changes in the intensity of sampling through time, or radiocarbon dating inaccuracies.

Biotic homogenization has been observed for a number of modern assemblages. For instance, heavily invaded aquatic plant communities have increased in similarity by only ~0.02–0.05 (Jaccard's Index)[19], while island bird assemblages, which are more vulnerable to anthropogenic activities, have increased in mean similarity by ~0.04 (Jaccard's Index)[15,69]. The most dramatically homogenized modern floras and faunas include Brazilian forests and island mammal assemblages, which have increased in similarity by ~0.16–0.24 (median; Jaccard's Index)[70] and ~0.10 (median; Jaccard's Index; ranging from 0.00 to 0.40 with only a very small proportion of islands showing an increase greater than 0.2)[15]. Though similar to some island mammal

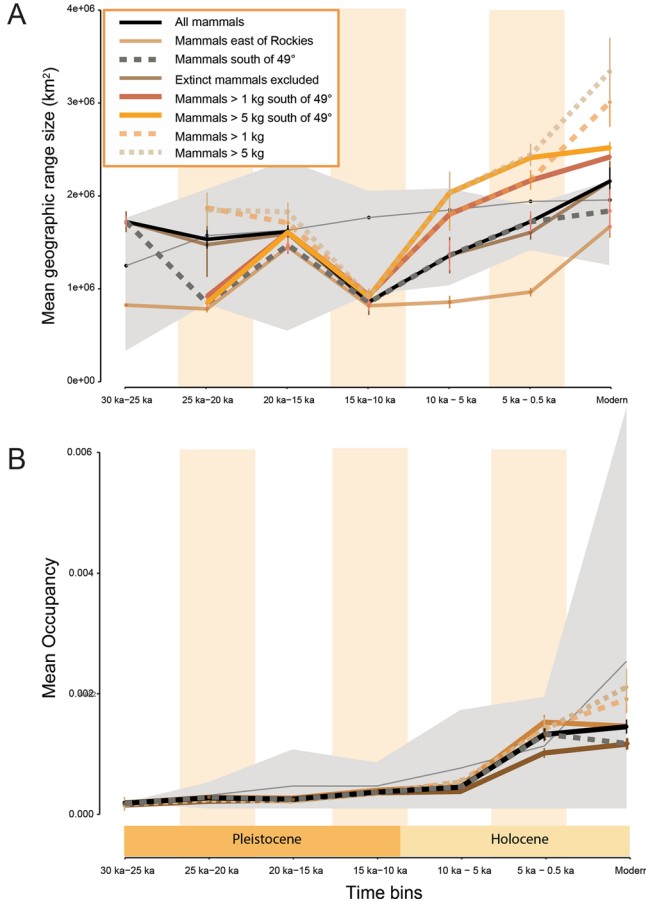

**Fig. 3 Mean mammal geographic range size increased from the 15,000–10,000 ybp time bin onward but only became significantly larger than null expectations in the modern time bin when small mammals (<1 kg) are included (sample sizes in Source Data).** Mean range size for mammals larger than 1 kg were significantly larger than null expectations by the 5,000-500 ybp time bin. Neither are associated with increases in range occupancy. **A** mean geographic range size ± standard error of the mean and **B** mean occupancy (proportion of 1° by 1° grid cells occupied) ± standard deviation. Gray ribbons show the mean of the null model runs (sites shuffled among time bins) with 95% confidence intervals.

assemblages, the magnitude of Pleistocene through Holocene biotic homogenization reported here (an increase of 0.15–0.25; >100% change in Jaccard similarity) exceeds that of modern and historical continental vertebrate faunas (typically < 0.05 on the Jaccard Index scale)[71,72]. These direct comparisons do not account for differences in time bin size (e.g., 5,000 years vs. years to decades), but do highlight the fact that more biotic homogenization has occurred among terrestrial North American mammal assemblages over the last 15,000 years than is captured by modern and historical studies alone.

To further contextualize our findings, we compared them to spatial patterns of modern Western Hemisphere mammal assemblage similarity using a space-for-time comparison[73]. Modern mammal assemblages are most homogenous at high latitudes and more heterogeneous at low latitudes, such as the Canadian Arctic versus central Mexico (Fig. 4)[66], a pattern that also has been documented elsewhere[66,74]. An increase in Jaccard similarity of ~0.15–0.25 (Fig. 1), as reported herein, is equivalent to the difference in mean assemblage similarity between the Arctic and subtropical faunas of the Western Hemisphere

(~30° of latitude) (Fig. 4). In other words, the change we observe over the past ~15,000–10,000 years is equivalent to what one would observe if the subtropical mammalian faunas of central Mexico became homogenized to the same extent as Arctic Alaskan faunas, which are currently more than twice as similar across comparable spatial distances (Fig. 4). Based on the above considerations, we are confident that we document biologically significant biotic homogenization of mammalian assemblages during the late Pleistocene and Holocene.

There are several potential drivers of late Quaternary biotic homogenization. Today, the comparative homogeneity of high latitude mammal faunas (Fig. 4) is attributed to lower climate heterogeneity (i.e., lower dissimilarity of climate among regions)[66,67]. Over millions of years, climate has also played a central role in structuring terrestrial mammalian faunas[75–77]; North American mammalian faunas over the last ~40 million years were more homogenous when climates were cooler and drier[78,79], suggesting the same may have been true for the late Quaternary. The ~30,000 year period of the present study encompasses the Last Glacial Maximum (LGM; ~21,000 ybp) and subsequent long-term warming and deglaciation of the Northern Hemisphere (19,000–11,000 ybp)[80], which is likely associated with weakening of latitudinal climate gradients[80,81] that could have affected the compositional similarity of mammal assemblages. The last ~19,000 years were also punctuated by several shorter periods of significant climate warming and cooling[81], including Heinrich Stadial 1 (17,000–14,700 ybp) and the Bølling-Allerød (14,700–12,900 ybp), which both coincided with glacial retreat[82–84]. During these periods, we expect decreased spatial heterogeneity of climate. The Younger Dryas (12,900–11,700 ybp) was a brief reversal of global warming and return to glacial-like conditions[83,85], during which we expect increased climate heterogeneity. Graham, et al.[47] implicated environmental changes in declining distance decay of similarity for North American mammal faunas between the Pleistocene and Holocene[81]. We show that climate heterogeneity declined primarily between 20,000 and 15,000 ybp, prior to the onset of biotic homogenization for mammals larger than 1 kg (Fig. 1). Further, mean similarity of mammalian assemblages could not be differentiated from null expectations until the 10,000–5,000 ybp time bin (Fig. 1), when there was comparatively little change in climate heterogeneity (Fig. 2), suggesting non-climatic drivers of biotic homogenization from late Pleistocene through Holocene (Supplementary Figure 1).

Biotic homogenization can also be driven by the extinction of species with narrow geographic ranges (Supplementary Fig. 1). The Pleistocene-Holocene transition (~11,700 ybp) was the culmination of the extinction of 72% of North American mammal genera larger than 44 kg, which began as early as 15,000 ybp but occurred primarily between 12,000 and 10,000 ybp[42,43,45]. The results included widespread range shifts[86], re-assembly of mammal communities[52,54], loss of functional diversity[87], and weakening of biotic interactions among surviving species[88]. Exclusion of the now extinct mammalian megafauna from our analysis, however, did not alter the observed pattern (Fig. 1; solid brown line), likely because the mammalian megafauna possessed geographic ranges that were similar in size to surviving taxa during the late Glacial and immediately prior to their extinction (Supplementary Fig. 9). All else being constant, the loss of the mammal megafauna by ~10,000 ybp should therefore not have favored homogenization or heterogenization (decreasing mean similarity among species assemblages; Supplementary Fig. 1). Excluding possible indirect ecological effects (e.g., trophic cascades), our analysis shows that the extinction of the mammalian megafauna did not directly lead to biotic homogenization.

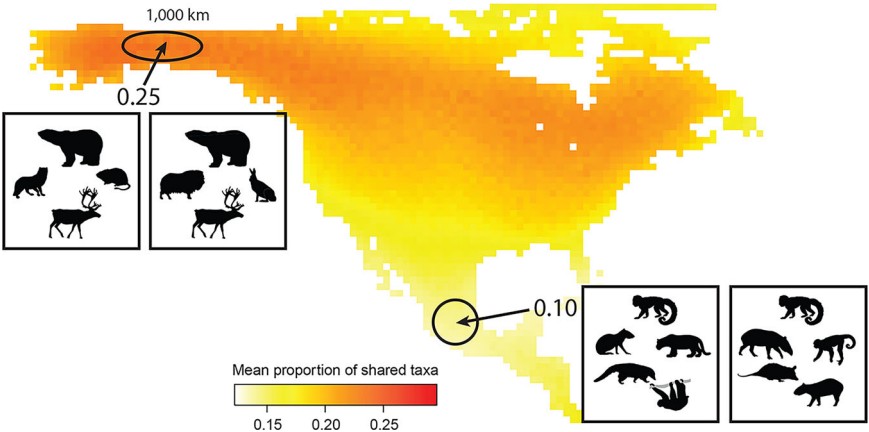

**Fig. 4 Latitudinal patterns of mean taxonomic similarity for modern Western Hemisphere mammals measured as mean proportion of shared taxa (Jaccard similarity) (sample sizes in Source Data).** An increase of 0.15-0.25 in assemblage similarity between ~15,000 ybp and Modern for North American mammals is equivalent to the difference in mean similarity between assemblages in Alaska and the same-sized regions in the subtropics at ~30˚ of latitude. The value of each cell is the mean Jaccard similarity of all surrounding cells within a 1000 km radius. Mean similarity between cells is 2.5× higher for Alaskan than for Mexican communities. Circles represent hypothetical assemblages of mammals from Arctic Alaska and Mexico. Silhouettes represent the occurrences of different species in adjacent grid cells. Silhouette credits from phylopic.org: Sarah Werning (*Bradypus*, Cebinae), Rebecca Groom (*Nasua*), Lukasiniho (*Panthera*). Silhouettes are distributed under Creative Commons Attribution 3.0 Unported license.

Nonetheless, North American mammal assemblages were significantly homogenized (i.e., outside the confidence intervals for the null model) by the 10,000 ybp–5,000 ybp (mammals larger than 1 kg) and 5,000–500 ybp time bins (all mammals) (Fig. 1; Table 1). We suggest a connection to the extinction of the majority of the megafauna that occurred between 12,000 and 10,000 ybp[45] (Fig. 1; star symbol). Although the loss of much of the North American megafauna did not directly drive Holocene biotic homogenization (Fig. 1; solid brown line), it may have done so via indirect ecological effects. Because most of the now extinct megafauna became extinct near the 10,000 ybp boundary[45], we suggest that biotic homogenization began late in the 15,000 ybp–10,000 ybp interval (mammals larger than 1 kg) and do not posit a long delay between the extinctions and their ecological effects.

Large mammals perform a variety of ecological functions, including facilitation of seed dispersal over long distances[89,90], maintenance of vegetation structure at the landscape-scale[91], moderation of small mammal populations through competition and predation, and, perhaps most importantly, lateral transfer of nutrients[92]. The causes for the global megafaunal extinctions are still debated and there is conflicting evidence regarding the direct or indirect role played by humans[42,44,93,94]. Regardless of causal mechanisms, removal of ecosystem engineering megafauna and the resulting dominance of smaller-bodied mammals with different functional roles[42,87] had continental-scale ecological consequences for North American terrestrial ecosystems[95–97], including geographic range expansions and shifts among surviving species[53,88].

Today, geographic range expansion, whether climate-mediated or via translocation, is one of the primary drivers of biotic homogenization[14,19,71] (Supplementary Fig. 1). After the 10,000–5,000 ybp time bin, we observe ~50% increases in average geographic range size among mammals larger than 1 kg (Fig. 3A) but not increases in range fill (Fig. 3B), consistent with previous studies of mammal range dynamics[86,88]. These range expansions exceeded the increase in newly available space and equitable climate resulting from glacial retreat alone[88], suggesting additional, possibly ecological, drivers. One possible ecological driver, landscape-scale ecosystem changes resulting from the loss of the megafauna, may have driven biotic homogenization for assemblages of mammals larger than 1 kg, given their close temporal association (Fig. 1). Assemblages including smaller mammals (<1 kg), however, showed much more moderate and delayed

increases in geographic range size and biotic homogenization until after ~5,000 ybp (Figs. 1 and 3A), possibly reflecting differences in dispersal abilities[98].

Regardless of the taxonomic or spatial filters we applied to the data, the fastest rate of biotic homogenization occurred between the 5,000–500 ybp and modern time bins (Fig. 1), coincident with geographic range expansions of ~25% for assemblages including all mammals (Fig. 3A). This second phase of biotic homogenization began during enhanced fire regimes, considerable human population growth, and the development of extensive agriculture (i.e., non-continuous but widespread cultivation) in North America[3,37]. Human populations may have increased by as much as 10-fold in North America during the penultimate time bin (5,000–500 ybp)[3]. Habitat alteration by human activities (e.g., clearing of forests, construction of villages) favors population growth among synanthropic species (i.e., those dependent on human-dominated habitats). Such species are favored due to their reliance on resources provided by human habitation (e.g., refuse) and the elimination of their natural predators (e.g., through hunting), among other factors. As human populations and habitation become more widespread, so do synanthropic species, leading to biotic homogenization[99]. Though it is likely such processes were operating during the mid to late Holocene (the 5,000–500 ybp time bin), they are very unlikely to have been as spatially extensive or intensive as is observed for modern urban environments. The incidence of fires also increased throughout the Holocene in North America, though, in most cases, this has been linked to climate change rather than anthropogenic activities[3]. Furthermore, fires may in fact produce less homogenous biotas[100].

Archeological evidence, however, suggests that, by ~2,000–1,000 ybp (within the penultimate time bin of this study), extensive agriculture was practiced throughout much of the central and eastern United States[3,37]. Modern agriculture results in spatially-extensive monocultures[101], favoring agricultural pests (e.g., voles)[102] and generalist species[103,104]. Today, agricultural intensification is one of the major drivers of biotic homogenization, largely due to reduced landscape heterogeneity and increased patch size[23]. Similarly, agriculture reduces the capacity of ecosystems to support large pools of species, resulting in spatially homogenous, depauperate floras and faunas[105]. Early farming practices did not produce the spatially extensive, single-species monocultures of today[106,107]. Our findings suggest,

however, that some combination of anthropogenic processes, which occurred 2,000–1,000 years ago (e.g., habitation, agriculture), may still have had a homogenizing effects on North America's land mammals (Fig. 1)[101].

Modern North America is characterized by the most homogenous mammalian faunas of the last 30,000 years (Fig. 1; Table 1). The 20th century saw the fastest and most spatially extensive human landscape modification, including the development of true agricultural monocultures, further transformation of the landscape into farmland, expansion of human transportation networks, and rapid growth of urban environments[2,6]. We suggest that modern human niche construction (e.g., agriculture) resulted in a dramatic intensification of ongoing biotic homogenization, producing a further increase of ~0.10 in faunal similarity (67% increase; Jaccard similarity; Fig. 1). Because much of our modern data is based on surveys from state and national parks, our results further suggest that the homogenizing effects of landscape modification have percolated through protected ecosystems. Assuming current rates of invasion and human landscape modification are sustained, North American mammals are projected to homogenize by a further 0.05–0.12 (Jaccard's Index) during the 21st century, given current human population densities and rates of species endangerment[108]. If these projections are borne out, continental North American mammalian assemblages will have become as much as 0.37 (Jaccard similarity) more homogenous over the last 15,000 years, an increase in similarity of greater than 300%. For context, such an increase would exceed the difference between modern Arctic and subtropical faunas (Fig. 4).

The scale of projected future biotic homogenization is of concern to wildlife managers, conservation biologists, and human communities relying on a variety of ecosystem services. Such extreme homogenization signals the replacement of complex and spatially variable ecosystems by a few, widespread, simpler ones[22], potentially driving the loss of resilience to perturbation[109]. Reduced resilience may result from the loss of functional redundancy, reduction in "response diversity", and, thus, the probability of species surviving future perturbations[110]. The ultimate consequence may be the loss of important ecosystem functions (defined as the transfer of energy and matter among individual species in a community)[26]. Continuing biotic homogenization into the future may have further consequences for the trajectories of biodiversity, as species-environment relationships are uncoupled, including our ability to forecast ecosystem-scale biodiversity changes, and, more fundamentally, to understand the drivers of biodiversity patterns.

We report biotic homogenization for modern continental North American mammal faunas and incorporate timescales allowing comparison of pre- or low human impact faunas with strongly anthropogenically-impacted ones. We show that biotic homogenization is not just a recent historical phenomenon, it preceded the modern era by as much as 10,000 years. We suggest that Holocene biotic homogenization proceeded in two phases, the first following the extinction of the mammalian megafauna, primarily impacting assemblages of mammals larger than 1 kg, and the second, coinciding with the development of extensive agricultural practices in North America and the rapid spread of anthropogenic biomes that now characterize much of our planet (Fig. 1)[111]. Our findings therefore contribute to ongoing discussion regarding the long-term environmental impacts of humans and the beginning of the Anthropocene[37,112] by showing that biotic homogenization began thousands of years before present.

## Methods
A diagram summarizing the various methods is included in Supplementary Fig. 2. All R code is available in the Github Repository[113].

**Fossil and modern data**. North American fossil mammal records are well resolved taxonomically and chronologically, making mammals ideal for understanding the effects of late Quaternary anthropogenic and climatic perturbation. Lists of species at individual sites are taken from a dataset containing 67 extant communities[114,115], as well as 298 Late Pleistocene and Holocene communities. Extant species occurrences are derived from a variety of literature and web sources and reflect occurrences during the 1970s to 2000s (operationally modern for the purposes of the present study)[114,115]. The Late Pleistocene and Holocene occurrences are derived from the FAUNMAP II database (2003), currently the most comprehensive compilation of North American Quaternary mammal occurrences that includes information about taxonomy, geochronology, geology, and taphonomy[116]. We required each assemblage of species to meet specific criteria for inclusion into the data set. First, assemblages were included only if they comprised a minimum of 20 species, based on sites with the lowest species richness included in ref. [114]. Second, assemblages were required to include at least one occurrence of species belonging to each of Rodentia, Artiodactyla, and Carnivora, assuring representation from the clades comprising the majority of non-volant North American mammal diversity.

**Delineating and dating fossil sites**. We defined sites from the FAUNMAP II database as spatially and/or temporally separated occurrences of mammal species meeting the richness and taxonomic criteria listed above. We included fossil sites that have been dated using both radiocarbon and biostratigraphy. Radiocarbon dates were calibrated using the BchronCalibrate function in the Bchron R package (v. 4.7.5)[117] using the IntCal13 calibration curve. We considered two or more stratigraphic layers within the same deposit (sharing the same latitude and longitude), but with calibrated radiocarbon ages that differed by more than 500 years, as separate sites. Layers within that same deposit that were dated to be within 500 years of each other were combined for the purpose of increasing the number of usable sites. Combining dated layers increases time-averaging but is unlikely to influence temporal trends, given that sites were then apportioned to longer time bins (as below). Similarly, two layers with different latitudes and longitudes but similar or identical radiocarbon ages were considered separate sites. We plotted the sites on maps for each time bin using the sp (v. 1.4.5)[118], raster (v. 3.4.10)[119], mapdata (v. 2.3.0)[120], maps (v. 3.4.0)[121], and ggplot2 (v. 3.3.5)[122] R packages (Supplementary Fig. 10).

Inaccuracies in bulk radiocarbon dates (those dating samples that have been performed prior to the use of Accelerator Mass Spectrometry for radiocarbon dating) are commonly as large as 2,000 years[123]. Furthermore, a number of sites in the dataset have no associated radiocarbon dates and are dated based on biostratigraphic methods. Thus, to be conservative and avoid false precision, we use 5,000-year time bins. Fossil sites were therefore divided into six 5,000 year time intervals (larger than the typical bulk radiocarbon dating error) based on their associated median radiocarbon or biostratigraphic dates: 30,000–25,000 cal. BP, 25,000–20,000 cal. BP, 20,000–15,000 cal. BP, 15,000–10,000 cal. BP, 10,000–5,000 cal. BP, and 5,000–500 cal. BP. As a test of sensitivity of 5,000 year bins with the above cut off times, we also used a series of bins that aligned with the end of the beginning of Heinrich Stadial 1 (17,000–14,700 ybp), end of the Younger Dryas (12,900–11,700 ybp), and the end of the mid Holocene (8,326–4,200 ybp). The resolution of the mammal data is such that it cannot be divided more finely without adding false precision.

We created the following subsets of the data: (1) only sites occurring east of the Rocky Mountains, thus excluding the most topographically complex regions of North America, which are associated with high rates of community turnover[66], (2) only sites occurring south of the Canada/USA border, given that few sites exist north of the border until the most recent time bins due to glacial coverage (Supplementary Fig. 10), (3) all sites excluding extinct megafauna, (4) all sites including species >1 kg, and (5) all sites including species > 5 kg, because the preservation potential, fieldwork collection practices, and dispersal abilities of small and large mammals differ[63,98].

**Measures of taxonomic similarity**. We use the term biotic homogenization to refer specifically to the increasing average compositional similarity of samples (i.e., a decrease in β diversity) through time[124]. All analyses were performed in R version 4.03[125] using the R packages betapart (v. 1.5.4)[126], vegan (v. 2.5.7)[127], fossil (v. 0.4.0)[128], and glm2 (v. 1.2.1)[129]. The Forbes similarity index was calculated using code by J. Alroy (http://bio.mq.edu.au/~jalroy/Forbes.R)[60]. To test for biotic homogenization, we used several metrics for compositional similarity and turnover, including the average of each of two measures of pairwise site similarity, the Jaccard and corrected Forbes similarity indices[60]. We also partition β diversity into its turnover and nestedness components using Simpson similarity (for turnover) and the difference between Sørensen and Simpson similarities (for nestedness) in their pairwise and multi-site implementations[130]. We also employed two additional methods, Distance Decay of Similarity and Multivariate Dispersion[131,132]. We include these various metrics for completeness. However, studies of biotic homogenization typically employ the Jaccard index[58].

Among the multitude of available similarity metrics, the Jaccard Index is preferred for calculating compositional similarity among samples, given that it excludes joint absences[133]. The corrected Forbes Index, also excludes joint absences but is intended to better compensate for the known relationship between within

site richness and values of the Simpson, Sørensen, and Jaccard Indices[60]. We also employ Baselga's method of partitioning nestedness (i.e., when smaller samples are subsets of larger samples) and turnover (i.e., replacement of species among samples), which are both components of β diversity. Because the Simpson Index measures turnover without the influence of richness, the difference between the Sørensen and Simpson indices can be taken as a measure of nestedness[130]. We also employ the multisite versions of these indices here but note that they are not easily comparable among time bins with different numbers of sites, and they produce biased estimates when the number of communities is unknown (i.e., when the number of sampling units may not equal the number of sites)[130,134]. Fortunately, pairwise metrics, as employed herein, circumvent these issues[134].

Distance decay of similarity is a measure of the steepness of the slope of a regression of similarity calculated using a pairwise dissimilarity metric, in this case the Jaccard Index, as a function of distance[132]. Multivariate dispersion measures differences in compositional dissimilarity among areas, using mean distances from the centroid of a principal coordinates analysis of dissimilarity matrices calculated using metrics such as Jaccard. Multivariate dispersion facilitates comparison among areas or groups[131]. It has not, however, been widely applied to study biotic homogenization so is not easily comparable to existing studies.

**Range size and occupancy**. Range sizes and occupancy were calculated from the fossil occurrence data. To delineate the proximate drivers of biotic homogenization, we calculated geographic range size (km$^2$) for each mammal species occurring at five or more sites in each time bin with multiple 95% convex hulls using a resampling approach implemented in the adehabitatHR R package (v. 0.4.19)[135]. The method involves estimating the geographic range using single-linkage cluster analysis (clusthr function) followed by size estimation at the 95% level (MCHu2hrsize function). We then calculated the mean geographic range size for each time bin.

To calculate mean proportional occupancy, we divided North America into 100 km by 100 km grid cells under a Behrmann equal area projection (9196 cells). We then assigned sites to a grid cell based on their proximity to its center using the R function spDistsN1in the sp R package (v. 1.4.5)[118], thus creating a species by grid cell occurrence matrix. We used this method because sites clustered in space are not treated individually and, thus, occupancy is not inflated. For each species occurring in each time bin, we calculated the proportion of grid cells occupied as a proportion of the total number of cells. We then calculated the mean proportional occupancy for each time bin.

**Null model comparison**. We test whether mean taxonomic similarity, range size, and occupancy vary across the seven time bins. However, simple regression analyses of similarity versus time is problematic for at least two reasons. First, the number of time bins ($n = 7$) is too small to provide credible results for a simple parametric correlation analysis, particularly if there are influential points that dominate the regression. The second problem is that biodiversity metrics, including species similarity, can be sensitive to various measures of sampling intensity, including the number of samples, the number of species, the occupancy or fill of the matrix, area samples, and the distance among samples (an issue due to spatial autocorrelation)[136]. We therefore used null model analyses to address these problems. As an alternative to the simple regression analysis, we used a null model that randomizes the assignment of each site to a particular time bin. This null model was first introduced for analyses of species co-occurrence[137]. It preserves the species association within each site, the total number of species occurrences across the time span, and the number of sites per time bin. However, it reshuffles patterns of site associations that change through time (shuffling among bins). Operationally, the null model involves drawing sites with their observed complement of species randomly from the entire pool (i.e., all sites across all time bins) without replacement until the number of observed sites for each time bin is achieved. Mean Jaccard similarity, geographic range size, and occupancy were then calculated for each time bin. The procedure was repeated 1,000 times. The null model employed herein allowed us to test for changes in species similarity through time and to test for changes in geographic range size and occupancy.

To assess the significance of observed changes in compositional similarity among sites, we calculated effect sizes as: (mean observed–mean null)/standard deviation of null. We interpreted effect sizes greater than zero as representing biotic homogenization. As an additional control, we used a regression analysis of Jaccard similarity and richness to ensure that there was no statistical correlation with the number of species recorded in each site, which often affects similarity measures.

**Radiocarbon ages**

As mentioned, errors in bulk radiocarbon dates pre-dating the application of Accelerator Mass Spectrometry methods are commonly as large as 2,000 years[123]. Therefore, site ages determined using bulk radiocarbon dating may be assigned incorrectly to a time bin. To test for the effects of radiocarbon dating inaccuracies, we used a randomization approach. Error was added to dates using random draws from a normal distribution with a mean of zero and standard deviation of 2,000 years. Mean taxonomic similarity was then re-calculated for each time bin. The procedure was repeated 1,000 times. For the radiocarbon error analyses, we calculated mean taxonomic similarity using the Jaccard index.

**Re-sampling**. The intensity with which the fossil record is sampled is inconsistent through time or space[138–141]. Such changes in sampling (e.g., the number of sites and the distances among them) can affect measures of taxonomic similarity and quantification of other macroecological phenomena. To assess the effects of sampling intensity and area on our analyses of taxonomic similarity, we first used a re-sampling approach in which we iteratively selected 15 sites per time bin (the number of sites in the poorest sampled time bin) and re-calculated taxonomic similarity 1,000 times for each dataset. We then determined the minimum longitudinal extent (i.e., the maximum longitudinal distance between sites for the time period with the smallest longitudinal spatial coverage) for the entire dataset (all sites in all time bins) and used the same approach while simultaneously limiting the longitudinal extent of the data for each of the 1,000 iterations. For the sampling intensity analyses, we calculated mean taxonomic similarity using the Jaccard index. This approach therefore accounts for changes in the number of sites as well as their density (mean distance among sites) during the sampling period.

**Space-for-time comparison**. The mean pairwise taxonomic similarity of mammal assemblages is extensively studied in modern contexts on a variety of spatial scales, particularly in the Western Hemisphere[66,142]. To contextualize the patterns observed in the fossil record and as a means of assessing whether observed changes may be biologically significant, we make a space-for-time comparison[73]. We downloaded spatially referenced geographic range data for modern non-volant Western Hemisphere mammals[143], a dataset that uses the taxonomy of ref. [144] and includes 1,366 species. The Western Hemisphere mammal dataset has been used in other recent studies of community structure[145]. We sampled the ranges of extant Western Hemisphere mammals using a Behrmann equal area projection and 100 km by 100 km grid cells using the R packages raster (v. 3.4.10)[119] and maptools (v. 1.1.1)[146] because smaller grid cell sizes are more subject to false positives (more likely to record a species as present when it is not)[147]. We considered grid cells to be occupied by a species if the center of the cell intersected with its geographic range[145]. The result was a species by grid cell occurrence matrix, which we used for further analyses.

We used the R function spDistsN1 in the sp R package[148] to calculate the great circle distances amongst grid cells. Using a spatial window of 1000 km, we subsampled grid cells surrounding each focal grid cell using the inverse of the great circle distance as the probability of selection[66]. For each subsampled group of grid cells, we calculated taxonomic similarity using the Jaccard Index. We then plotted the values back onto projected maps of the Western Hemisphere under a Behrmann equal-area projection.

**Paleoclimate data**. To test for an association between climate heterogeneity and mean mammal taxonomic similarity, we downloaded North American paleoclimate data from de-biased and downscaled earth system model (ESM) climate simulations based on recent paleoclimate models at 0.5 degree resolution[68]. We downloaded four climate variables from the Dryad digital database (http://datadryad.org/resource/doi:10.5061/dryad.1597g): annual average minimum temperature, annual average maximum temperature, annual actual evapotranspiration (AET), and total annual precipitation. These climate variables are known to correlate with mammal diversity[67,149]. Climate variables of 500-year intervals were a) sampled evenly across the continental USA and southern Canada, where the majority of mammal fossil sites are located and b) averaged into 5,000-year time intervals for comparison to fossil mammal data. Climate and mammal data were projected into a WGS-84 global coordinate system for analysis.

We summarized the paleoclimate data using principal components analysis (PCA). We subjected the paleoclimate data for all sites in each time slice to a single PCA and extracted values for the first and second principal components. We then calculated climate turnover as the mean pairwise climate dissimilarity in PCA scores amongst sites in each time slice.

**Reporting summary**. Further information on research design is available in the Nature Research Reporting Summary linked to this article.

## Data availability

Pleistocene and Holocene mammal data were collected from the published Faunmap 2.0 database (https://ucmp.berkeley.edu/faunmap/). Data from individual fossil and archeological sites were not collected by the authors of the present study. Faunmap was assembled from published occurrence data (described here: https://ucmp.berkeley.edu/faunmap/). Modern mammal occurrence data were collected from the following sources, Brown & Nicoletto (1991) and Lyons & Smith (2013). Brown & Nicoletto (1991) and Lyons & Smith (2013) compiled mammal occurrence data from various sources, including within national parks, as described therein. Software used in data collection included an internet browser (Google Chrome) and Microsoft excel. The data used in this study have been deposited in the Github repository https://doi.org/10.5281/zenodo.6518845. Source data are provided in this paper.

## Code availability

The R code used during the current study is available in the Github repository, https://doi.org/10.5281/zenodo.6518845.

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

## Acknowledgements
D.F. works on the traditional and unceded territories of the Algonquin-Anishnaabe Nation. This work was supported by the Smithsonian Institution's National Museum of Natural History Program grant to the Evolution of Terrestrial Ecosystems Program (ETE) and by a NSF Research Coordination Network Grant DEB 1257625 to S.K.L., A.K.B., and N.J.G. This is ETE publication #352. DF was supported by a Discovery Grant from the Natural Sciences and Engineering Research Council of Canada, Research Activity Grants from the Canadian Museum of Nature, and a Peter Buck Postdoctoral Fellowship endowed by Dr. Peter Buck and awarded by the Smithsonian's National Museum of Natural History.

## Author contributions
All authors (D.F., A.V., A.B.T., M.A.B., J.T.E., W.A.B., A.K.B., M.D., A.D., J.T.F., G.R.G., N.J.G., A.M.J., C.V.L., B.J.M., J.H.M., S.P.M., R.P., A.B.S., L.C.S., S.K.L.) contributed to the development and interpretation of the ideas in this manuscript and edited the writing. D.F. curated and analyzed the data, produced the figures, and wrote the paper. A.V. performed part of the data analysis, aided in the writing of early versions of the manuscript, and aided in figure design. A.B.T. provided R code for data analysis and analyzed early versions of the data. M.A.B. aided in figure design and in the writing of early versions of the manuscript. J.T.E. aided in the writing of early and recent versions of the manuscript.

## Competing interests
The authors declare no competing interests.
