## [Peer Review File · Nature Communications]

LATE QUATERNARY BIOTIC HOMOGENIZATION OF NORTH AMERICAN MAMMALIAN FAUNASREVIEWER COMMENTS

Reviewer #1 (Remarks to the Author):

The authors compile a suite of contemporary and paleo faunal databases to assess the homogeneity of mammalian communities in North America over a period of time that includes pre-anthropogenic effects. They show that homogenization is not a recent phenomenon, but one that has likely resulted – directly or indirectly – from anthropogenic effects over many thousands of years. I enjoyed reading this paper, but I also found it very much a black box... for lack of a better term.

Nowhere is it detailed how exactly these community measures are made other than to list a suite of similarity metrics... In other words, it is very important to ensure that the reader understands how a) taxonomic similarity, b) range sizes/occupancy, and c) those same measures as used by the null model, are pieced together to obtain the described results. I found this very difficult to follow, because there is no explicit (and straightforward) description of how and what data flows together to derive the effect sizes. Some equations and/or a diagram illustrating what data are used and how would be very helpful. To this end, there is much data-wrangling that the authors employ (and I am not asserting that it is not justified) — for example to get estimates of species ranges — but these steps are really impossible to evaluate without the authors making the dataset and code that they are using available.

Here is the biggest issue that I have with this contribution: because the data are gathered from so many sources (it is not clear if all paleo data are published elsewhere?), and wrangled in such a variety of ways, it would be nearly impossible to reconstruct the author's analyses without an incredible effort and much guesswork. There are a number of integrated calculations using functions from different software packages, and it's difficult if not impossible to get a sense of whether species/site calculations using these packages are reasonable, and we are left to trust the authors. To this end, too much of the current manuscript is based on verbal explanations of their methods without explicit descriptions of their calculations, or a means by which to evaluate whether these calculations are reasonable. To this end, I think it is reasonable to expect that the author's make their code and the curated data used in this paper directly available so that the assumptions behind the many data transformations can be directly evaluated by the community. My understanding is that this is a pre-requisite for Nature journals?

(the above criticism assumes that the code/data are not available... at least they are not included in the submission material that I have, so I am making the assumption this is the case. If I'm in error, of course this criticism isn't valid)

Specific comments

L90-92: perhaps the point could be made more strongly here... that keystone species - whose impact is inversely proportional to their relative abundance - are being lost and replaced with dominant species, where impact *is* proportional to their relative abundance. In other words, in these homogenized communities there are still species who have large impact on ecosystem process, but they will tend to be those that are more common, in part because homogenized communities are losing those less common (but important) species

L100 – though one could argue that a 'human free' ecological baseline is not enormously meaningful in the Holocene/Anthropocene

L111 – not all of these references directly examine the effects of changes in biotic interactions (refs 51-54). Consider including citations of some paleo interaction web literature, where the effects of interactions are directly assessed (for example changes in pollination webs and trophic webs in response to anthropogenic pressures in Pires Ecography 2018, Pires et al. ProcRoySoc 2015, Galetti et al. Biol Rev 2018, etc.) [note: this reviewer is not an author in suggested citations]

L124 – is this a complete description of the vetting process? The wording makes it unclear. If it is not

complete, I would suggest including a complete list of the criteria (or citing where to find a complete list) — coming back to this after reading the manuscript, I think it is complete, so consider changing the wording to emphasize this

L149 – While the details of the null model deservedly go in the methods, I think some clarification of the utility of this specific null model is warranted here. Also, is there utility to consider a null model generated by shuffling time-bins within sites? While a shuffled sites within timebins gives you a sense of the significance of measured homogeneity from site to site, would a shuffled times within sites null give you a sense of the significance in homogeneity differences from one time bin to another? Or perhaps $n=7$ makes such a model too difficult to interpret

L153 – the format of times needs to be consistent... early in the ms, xx ka is used... here it switches to 5000 yrs bp

L168 – I think the reader needs to be given a sense of how large differences in topographic heterogeneity or trends in sampling area/density are to get a sense of how these modified results account for such potential biases (or at least pointed to references that do?). I'm having a hard time getting a sense of what excluding west of the Rockies or north of the Canadian border are meant to achieve without a better description of the problem these altered results are supposed to surmount.

L381-390 Are these different measures of similarity discussed anywhere? It seems strange to present the results in the supplement, without any discussion for the positives/negatives/assumptions of the different metrics. Also shouldn't this section link to the supplemental fig?

L392 "Range sizes and occupancy were calculated from the data described above, not the mammal distribution data described below" — this is not a very clear statement... I'm not sure how to interpret this.

L424-426 Again, mentioning regression analyses without specifying what exactly is being regressed, the summary stats, or pointing to a figure/table is not very helpful

Reviewer #2 (Remarks to the Author):

This is an interesting and well written paper that uses current and prehistoric data from North America to explore the impact of humans and climate (and other factors) on the similarity in species composition among ecological communities. I think this paper is an excellent contribution to the general discussion about the direct and indirect impact of humans on the ecological communities. Additionally, the impact of the paper increase because the observational design include late Pleistocene and Holocene communities, as well as due to the authors explore the effect of late Pleistocene human arrives and Holocene agriculture origin. For these reasons I recommend the publication of the manuscript after minor review.

I have three comments:

1- I am not specialist in North American faunal assemblages, but I see that the authors do not perform an original bibliography review, but rather use online databases from North America. I believe that this is the normal procedure in the studies carried out in North America, but I recommend a short discussion of the quality of the dataset.

2- The authors use equal time intervals (5 ka) to analyze changes in communities in the last 30ka. However, because the authors are interesting in climate factors influencing the changes in composition among ecological communities, they could they could have used "natural" intervals, such as late-Holocene (4.2-0.3 ka), mid-Holocene (8.326-4.2 ka), early-Holocene (11.7-8.326 ka), Younger Dryas Stadial (12.9-11.7 ka), Bølling-Allerød (14.7-12.9 ka), Heinrich Stadial 1 (17.0-14.7 ka)

(<http://www.paleoclim.org/>), which correspond with recent climatic variations [Fordham et al. 2017. (<https://doi.org/10.1111/ecog.03031>); Brown et al. 2018 (<https://doi.org/10.1038/sdata.2018.254>)]. Please, explain the reasons for the methodological choice.

3- Just like any other phenomenon in nature, the measures of similarity among ecological communities (e.g. Jaccard similarity) are influenced by spatial and temporal auto-correlation. Did the authors consider this aspect in the analyses? I can not find this information in the description of methods used to explore temporal and spatial changes. This could be important to consider the robustness of the conclusion in relation to the main factors (human arrives and agriculture origin) hypothesized as the causes of the observed changes.

Best regards

S. Ivan Perez

Reviewer #3 (Remarks to the Author):

This is a carefully thought out and well written paper, with a range of considered analyses that builds on previous work on the loss of mammal communities in North America from the late Quaternary to the present. The figures are clearly put together and easy to interpret. Well done to the authors. I don't have any substantive issues with the paper, just a few comments on the presentation of methods and interpretation of results that it would be helpful for the authors to address within the manuscript.

1. Figures 2 and 3 seem to be referred to the wrong way around several times in the manuscript e.g. p. 9, 11, 12 & 15 but maybe elsewhere too.

2. A brief discussion of diversity indices in your Introduction would help clarify your approach for the reader e.g. alpha vs beta diversity; why you are interested in beta; how that can be measured. You also need to actually explain what the Jaccard similarity measure is, ideally both in the Introduction briefly and then in more detail in the Methods, including what it measures and how, as well as explaining the actual index i.e. it's on a scale of 0-1, with 1 being more homogenous etc.

Given you go to the trouble of comparing several different measures it would be good to explain the divergent results you find in two of your measures (the 'nestedness' and 'Forbes' lines in Fig. S2), which show predominantly downward trends, at least until the most recent time bin, contrary to your main results. Why might this be? In the methods you also refer to the various indices using different names than in Fig S2 – could these be aligned to make it clearer?

3. In assessing climate as a driver, we are generally interested in the limits of species tolerances to climate rather than their average conditions. Therefore ideally annual minimum and maximum temperature should be used rather than annual average of the minimum/maximum (if that is available, I am not familiar with the particular climate dataset but looks similar to one I have used).

On p.12, line 228-9 'lower climate heterogeneity (i.e. lower dissimilarity of climate among regions)' is rather confusingly phrased! Could 'lower dissimilarity' be changed to something like 'increased similarity' or 'more similar climate' as that is instinctively a little easier to understand?

I am rather unconvinced as to how your climate analysis is allowing you to fully test climate as a driver. The time periods you consider incorporate huge climatic variability in North America (LGM, Younger Dryas, Holocene warming), including substantial biome shifts from steppe-tundra around 15ka (see Allen et al. 2020 <https://doi.org/10.1111/jbi.13930>) and the massive ice sheet retreat. Some of these rapid changes were happening on centennial to millennial time scales (see e.g. <https://openquaternary.com/articles/10.5334/oq.46/>). Therefore picking climate at 500 yr intervals and averaging into 5000 yr time bins is likely to mask environmental transitions that we already know

precipitated faunal turnover and changes in species composition. Indeed, surprisingly your analysis does not find 15-10ka to be a period of high climate heterogeneity. Can the authors address this somewhat unexpected result?

Indeed, one of the first papers based on the FAUNMAP data (Graham et al. 1996) found similar results to the authors here in terms of increased faunal community homogenisation over the Pleistocene-Holocene boundary, but framed this explicitly in terms of response to known environmental transitions around this time. This is rather uncontroversial and can also be seen in Europe. In addition, the only evidence the authors provide for human influence is human arrival times, which is entirely correlative, so offers no more convincing argument than their coarse climate analysis. You would probably be better off comparing your results in Figure 1 against a highly resolved climate curve as a comparison with human drivers. As such, this part of the paper would benefit from a more rounded discussion of environmental and human drivers, including their probable interaction and acknowledging previous comparable papers such as Graham et al. more explicitly.

REVIEWER COMMENTS

We thank the reviewers for their helpful comments that have improved our manuscript. Please find our responses below.

Reviewer #1 (Remarks to the Author):

The authors compile a suite of contemporary and paleo faunal databases to assess the homogeneity of mammalian communities in North America over a period of time that includes pre-anthropogenic effects. They show that homogenization is not a recent phenomenon, but one that has likely resulted – directly or indirectly – from anthropogenic effects over many thousands of years. I enjoyed reading this paper, but I also found it very much a black box... for lack of a better term.

Nowhere is it detailed how exactly these community measures are made other than to list a suite of similarity metrics... In other words, it is very important to ensure that the reader understands how a) taxonomic similarity, b) range sizes/occupancy, and c) those same measures as used by the null model, are pieced together to obtain the described results. I found this very difficult to follow, because there is no explicit (and straightforward) description of how and what data flows together to derive the effect sizes. Some equations and/or a diagram illustrating what data are used and how would be very helpful.

We have now created and include a “flow diagram” that we hope will help readers follow the various analyses used herein.

To this end, there is much data-wrangling that the authors employ (and I am not asserting that it is not justified) — for example to get estimates of species ranges — but these steps are really impossible to evaluate without the authors making the dataset and code that they are using available. Here is the biggest issue that I have with this contribution: because the data are gathered from so many sources (it is not clear if all paleo data are published elsewhere?), and wrangled in such a variety of ways, it would be nearly impossible to reconstruct the author’s analyses without an incredible effort and much guesswork. There are a number of integrated calculations using functions from different software packages, and it’s difficult if not impossible to get a sense of whether species/site calculations using these packages are reasonable, and we are left to trust the authors. To this end, too much of the current manuscript is based on verbal explanations of their methods without explicit descriptions of their calculations, or a means by which to evaluate whether these calculations are reasonable. To this end, I think it is reasonable to expect that the author’s make their code and the curated data used in this paper directly available so that the assumptions behind the many data transformations can be directly evaluated by the community. My understanding is that this is a pre-requisite for Nature journals?

(the above criticism assumes that the code/data are not available... at least they are not included in the submission material that I have, so I am making the assumption this is the case. If I’m in error, of course this criticism isn’t valid)

We have now made the analyzed dataset and code available on Github (https://github.com/danielleleefraser/biotic_homogenization). The link is provided on line 393.

Specific comments

L90-92: perhaps the point could be made more strongly here... that keystone species - whose impact is inversely proportional to their relative abundance - are being lost and replaced with dominant species, where impact *is* proportional to their relative abundance. In other words, in these homogenized communities there are still species who have large impact on ecosystem process, but they will tend to be those that are more common, in part because homogenized communities are losing those less common

(but important) species

The wording has been adjusted to indicate that the ecological impacts of keystone species are disproportional to their abundances.

L100 – though one could argue that a ‘human free’ ecological baseline is not enormously meaningful in the Holocene/Anthropocene

We have changed the wording to “low human impacts” when referring to the Pleistocene and early Holocene.

L111 – not all of these references directly examine the effects of changes in biotic interactions (refs 51-54). Consider including citations of some paleo interaction web literature, where the effects of interactions are directly assessed (for example changes in pollination webs and trophic webs in response to anthropogenic pressures in Pires Ecography 2018, Pires et al. ProcRoySoc 2015, Galetti et al. Biol Rev 2018, etc.) [note: this reviewer is not an author in suggested citations]

The suggested citations have been added.

L124 – is this a complete description of the vetting process? The wording makes it unclear. If it is not complete, I would suggest including a complete list of the criteria (or citing where to find a complete list) — coming back to this after reading the manuscript, I think it is complete, so consider changing the wording to emphasize this

Yes, it is. We have altered the text to make this clearer.

L149 – While the details of the null model deservedly go in the methods, I think some clarification of the utility of this specific null model is warranted here. Also, is there utility to consider a null model generated by shuffling time-bins within sites? While a shuffled sites within timebins gives you a sense of the significance of measured homogeneity from site to site, would a shuffled times within sites null give you a sense of the significance in homogeneity differences from one time bin to another? Or perhaps $n=7$ makes such a model too difficult to interpret

We have added some justification for the null model in the introduction.

The null model we use does not shuffle sites within time bins. It shuffles sites AMONG time bins. We have therefore already done exactly what the reviewer suggests. We have altered the text to make this clearer.

L153 – the format of times needs to be consistent... early in the ms, xx ka is used... here it switches to 5000 yrs bp

Fixed.

L168 – I think the reader needs to be given a sense of how large differences in topographic heterogeneity or trends in sampling area/density are to get a sense of how these modified results account for such potential biases (or at least pointed to references that do?). I’m having a hard time getting a sense of what excluding west of the Rockies or north of the Canadian border are meant to achieve without a better description of the problem these altered results are supposed to surmount.

We now more clearly explain that because the spatial coverage of sampling changes among time intervals and taxonomic similarity varies among latitudes and between lowlands and highlands, we must use sensitivity and resampling analyses to tease out the ecological signal.

L381-390 Are these different measures of similarity discussed anywhere? It seems strange to present the results in the supplement, without any discussion for the positives/negatives/assumptions of the different metrics. Also shouldn't this section link to the supplemental fig?

These are now discussed in the methods. It is not necessarily clear that one similarity metrics is categorically more appropriate than the others, except where we indicate so. However, the Jaccard Index is most often applied to studies of Biotic Homogenization, and we favor comparability.

L392 "Range sizes and occupancy were calculated from the data described above, not the mammal distribution data described below" — this is not a very clear statement... I'm not sure how to interpret this.

Fixed. This was meant to clarify that the range sizes were calculated from the fossil data not the modern distributional dataset, which is used only to create the map of modern North American mean taxonomic similarity among grid cells.

L424-426 Again, mentioning regression analyses without specifying what exactly is being regressed, the summary stats, or pointing to a figure/table is not very helpful

Fixed. We would like to avoid pointing to a results figure in the methods. The figure is referred to in the results.

Reviewer #2 (Remarks to the Author):

This is an interesting and well written paper that uses current and prehistoric data from North America to explore the impact of humans and climate (and other factors) on the similarity in species composition among ecological communities. I think this paper is an excellent contribution to the general discussion about the direct and indirect impact of humans on the ecological communities. Additionally, the impact of the paper increase because the observational design include late Pleistocene and Holocene communities, as well as due to the authors explore the effect of late Pleistocene human arrives and Holocene agriculture origin. For these reasons I recommend the publication of the manuscript after minor review.

I have three comments:

1- I am not specialist in North American faunal assemblages, but I see that the authors do not perform an original bibliography review, but rather use online databases from North America. I believe that this is the normal procedure in the studies carried out in North America, but I recommend a short discussion of the quality of the dataset.

The FAUNMAP II database is the most comprehensive compilation of North American Quaternary vertebrate localities. FAUNMAP complies locality information, taxonomic information, catalog numbers, NISP (number of identified specimens) and/or MNI (minimum number of individuals) when possible, geologic information, geochronologic information, and taphonomic information for fossil vertebrate (predominantly mammal) occurrences that range in age from about 5 million years through about 500 years old. The geographic scope covers the United States, Canada, and Mexico. It includes over 5000 localities and over 60,000 faunal occurrences. The first iteration of the database was published in 1994 with subsequent updates made in 2003. These data have now been incorporated into the widely used Neotoma Database as well. The first study to use the database was published in 1996 by Graham et al. and since then, it has become the commonly used source for North American Quaternary Data. FAUNMAP has gained wide acceptance within the paleontological community (see Uhen et al. 2013

JVP), and it is ubiquitous in Quaternary ecological studies (see Lyons 2003 J. Mammalogy, Schmitz et al. 2003 BioScience, Faith and Surovell 2009 PNAS, Waltari and Guralnick 2009 J. Biog., Barnosky et al. 2011 Geol. Soc. Lond. Spec. Pubs., McHorse et al. 2012 Paleo3, Pardi and Smith 2016 Ecography).

Given its wide acceptance and ubiquity, we argue that a citation to the original database [along with a slightly modified sentence] is sufficient to lead the reader to any further information that they might need, and requires no further discussion about the quality of the database itself. Furthermore, our working dataset includes a subset of FAUNMAP, and we have provided a clear description of the quality of those data, deposited those data in GitHub, and our code can be used to reproduce our analyses (https://github.com/danielleleefraser/biotic_homogenization).

2- The authors use equal time intervals (5 ka) to analyze changes in communities in the last 30ka. However, because the authors are interesting in climate factors influencing the changes in composition among ecological communities, they could they could have used "natural" intervals, such as late-Holocene (4.2-0.3 ka), mid-Holocene (8.326-4.2 ka), early-Holocene (11.7-8.326 ka), Younger Dryas Stadial (12.9-11.7 ka), Bølling-Allerød (14.7-12.9 ka), Heinrich Stadial 1 (17.0-14.7 ka) (<http://www.paleoclim.org/>), which correspond with recent climatic variations [Fordham et al. 2017. (<https://doi.org/10.1111/ecog.03031>); Brown et al. 2018 (<https://doi.org/10.1038/sdata.2018.254>)]. Please, explain the reasons for the methodological choice.

There are several significant issues with using smaller than 5,000 year time bins, as suggested here. The first, as mentioned in the text, is that most of the radiocarbon dates in Faunmap II are pre-AMS. It has been shown that pre-AMS dates can be inaccurate by up to 2,000 years (as cited and stated in the text). Furthermore, not all of the dates in Faunmap II are radiocarbon. Many are based on biostratigraphy, which adds considerably more uncertainty. We believe 5,000 year time bins are conservative and do not introduce false precision, as would be the case if we used smaller time bins.

Furthermore, species richness and turnover are known to decrease with increasing temporal grain (Adler et al., 2005; Tomašových & Kidwell, 2010). Therefore, using equally-sized time bins helps to control for this effect.

That being said, we take the point of aligning the time bins with the major climate events. However, due to the above-stated issues of dating precision, we cannot cut the data at all of the specified time points.

However, we were able to use bin cut off dates that align only with the end of the beginning and beginning of the Heinrich Stadial and end of the Younger Dryas (ending with the mid Holocene) (dates from above reviewer comment were used). The results are the same as with the 5,000 year time bins. We now include these as supplementary figures.

3- Just like any other phenomenon in nature, the measures of similarity among ecological communities (e.g. Jaccard similarity) are influenced by spatial and temporal auto-correlation. Did the authors consider this aspect in the analyses? I can not find this information in the description of methods used to explore temporal and spatial changes. This could be important to consider the robustness of the conclusion in relation to the main factors (human arrives and agriculture origin) hypothesized as the causes of the observed changes.

Spatial and temporal autocorrelation both describe patterns of greater similarity of samples that are closer together in either space or time. In practice, they lead to overestimates of effect sizes when regressing variables that are measured at different spatial or temporal points.

Our use of null models inherently controls for the effects of autocorrelation. The null model, by shuffling sites among time bins, also shuffles the spatial distribution of sites. So, the calculated effect sizes also

account for changes in the spatial distances among sites as well as the temporal distances among sites. We have tried to make this clearer in the materials and methods.

Best regards

S. Ivan Perez

Reviewer #3 (Remarks to the Author):

This is a carefully thought out and well written paper, with a range of considered analyses that builds on previous work on the loss of mammal communities in North America from the late Quaternary to the present. The figures are clearly put together and easy to interpret. Well done to the authors. I don't have any substantive issues with the paper, just a few comments on the presentation of methods and interpretation of results that it would be helpful for the authors to address within the manuscript.

1. Figures 2 and 3 seem to be referred to the wrong way around several times in the manuscript e.g. p. 9, 11, 12 & 15 but maybe elsewhere too.

Fixed.

2. A brief discussion of diversity indices in your Introduction would help clarify your approach for the reader e.g. alpha vs beta diversity; why you are interested in beta; how that can be measured.

It is our opinion that this belongs in the methods (where we have elaborated on this) and not in the introduction. We also believe that it is clear from the outset of the manuscript that biotic homogenization is a term that effectively means declining beta diversity.

You also need to actually explain what the Jaccard similarity measure is, ideally both in the Introduction briefly and then in more detail in the Methods, including what it measures and how, as well as explaining the actual index i.e. it's on a scale of 0-1, with 1 being more homogenous etc.

Fixed.

Given you go to the trouble of comparing several different measures it would be good to explain the divergent results you find in two of your measures (the 'nestedness' and 'Forbes' lines in Fig. S2), which show predominantly downward trends, at least until the most recent time bin, contrary to your main results. Why might this be? In the methods you also refer to the various indices using different names than in Fig S2 – could these be aligned to make it clearer?

The Forbes Index and Distance Decay metrics show clear correlations with within bin richness and number of sites, respectively. This is now explained in the text.

3. In assessing climate as a driver, we are generally interested in the limits of species tolerances to climate rather than their average conditions. Therefore ideally annual minimum and maximum temperature should be used rather than annual average of the minimum/maximum (if that is available, I am not familiar with the particular climate dataset but looks similar to one I have used).

The climate variables we use have all been shown to be related to mammal diversity. This is now clarified in the text.

On p.12, line 228-9 'lower climate heterogeneity (i.e. lower dissimilarity of climate among regions)' is

rather confusingly phrased! Could 'lower dissimilarity' be changed to something like 'increased similarity' or 'more similar climate' as that is instinctively a little easier to understand?

We prefer to continue using the original phrasing as "climate heterogeneity" and "climate turnover" are used elsewhere in the biodiversity literature.

I am rather unconvinced as to how your climate analysis is allowing you to fully test climate as a driver. The time periods you consider incorporate huge climatic variability in North America (LGM, Younger Dryas, Holocene warming), including substantial biome shifts from steppe-tundra around 15ka (see Allen et al. 2020 <https://doi.org/10.1111/jbi.13930>) and the massive ice sheet retreat. Some of these rapid changes were happening on centennial to millennial time scales (see e.g. <https://openquaternary.com/articles/10.5334/oq.46/>). Therefore picking climate at 500 yr intervals and averaging into 5000 yr time bins is likely to mask environmental transitions that we already know precipitated faunal turnover and changes in species composition. Indeed, surprisingly your analysis does not find 15-10ka to be a period of high climate heterogeneity. Can the authors address this somewhat unexpected result?

We have averaged climate over 5,000 year intervals so as to facilitate the comparison with the fossil data (please see below for an explanation of why smaller time intervals cannot be used).

We now include additional analyses including cut offs that align with some of the important climate events (see above). We also include an analysis of climate heterogeneity at 500 year intervals so as to show that we were in fact over estimating changed in heterogeneity rather than masking them, though the patterns are similar (see orange lines in Figure 3).

Given that 15-10 ka is a period of deglaciation, broadly, it is not clear to us that we would expect higher climate heterogeneity within the region sampled by the fossil sites, which is largely restricted to the continental USA and southern Canada. As such, our higher resolution plots do not show apparent increases in climate heterogeneity during the Younger Dryas, for example.

Copied from above:

There are several significant issues with using smaller than 5,000 year time bins, as suggested here. The first, as mentioned in the text, is that most of the radiocarbon dates in Faunmap II are pre-AMS. It has been shown that pre-AMS dates can be inaccurate by up to 2,000 years (as cited and stated in the text). Furthermore, not all of the dates in Faunmap II are radiocarbon. Many are based on biostratigraphy, which adds considerably more uncertainty. We believe 5,000 year time bins are conservative and do not introduce false precision, as would be the case if we used smaller time bins.

Furthermore, the climate data is sampled so as to replicate the spatial locations of the sampled fossil sites. The vast majority of the sites, as a result of the massive glaciation of North America, are found south of the USA/CAD border. Thus, the climate record, as summarized here, would not pick up changes in heterogeneity at higher latitudes (there are no fossil sites there following our strict vetting procedure except during certain time bins).

Additionally, we do see declining climate heterogeneity from the 20,000-15,000 time bin to the 15,000-10,000 time bin, which is entirely consistent with deglaciation. We would expect lower latitudes to become more homogenous, at least insofar as the steep climate gradient produced by the glaciers would have shifted northward, where we do not have any fossil samples.

Finally, there is no fruitful way to compare sites that cannot be partitioned amongst finer time bins to very finely resolved changes in climate.

Indeed, one of the first papers based on the FAUNMAP data (Graham et al. 1996) found similar results to the authors here in terms of increased faunal community homogenisation over the Pleistocene-Holocene

boundary, but framed this explicitly in terms of response to known environmental transitions around this time. This is rather uncontroversial and can also be seen in Europe.

Graham et al. (1996) perform their analyses using only two time bins, do not account for heterogeneity in spatiotemporal patterns of sampling, do not address differences between all mammals and large mammals, do not include a modern time bin, and employ a metric similar to distance decay of similarity, which is not easily comparable to historical and modern studies of biotic homogenization. They also implicate climate change without addressing whether changes in climate heterogeneity accompanied changes in beta diversity. Furthermore, we use the updated Faunmap II database.

In addition, the only evidence the authors provide for human influence is human arrival times, which is entirely correlative, so offers no more convincing argument than their coarse climate analysis. You would probably be better off comparing your results in Figure 1 against a highly resolved climate curve as a comparison with human drivers. As such, this part of the paper would benefit from a more rounded discussion of environmental and human drivers, including their probable interaction and acknowledging previous comparable papers such as Graham et al. more explicitly.

We now clarify how our study differs from and is a major expansion of the related work by Graham et al. (1996).

We now include more detailed discussion of patterns of climate change during the studied interval and address the role of humans a little more explicitly, though it is outside the scope of the present paper to weigh in on their role in the megafauna extinction.

REVIEWER COMMENTS

Reviewer #1 (Remarks to the Author):

Overall, I felt that the additions/edits made to the manuscript clarify the approach, and I commend the authors on their efforts during the revision. I still feel, though, that the link between the observed trends and stated drivers is a bit simplistic and at times over-reaches.

Specifically, the authors' interpretation of their homogenization data, and in particular their emphasis that homogenization began 15 Ka years before present, seems a bit misleading. To me, it is not immediately clear that the Pleistocene extinctions are the foremost culprit of the first noted increase in homogenization within the 10-5 Ka time bin (fig 1). As the authors state, the megafaunal extinctions occur within the 15-11.7 Ka window, yet homogenization does not appear increased during this period. It seems strange to me that there would not be any detected effects of these extinctions during the 15-10 Ka window if extinctions are the primary driver of homogenization... If extinctions are the sole cause, it implies that their effects are lagged enough that systems would not show their effects until thousands of years later... unless all of the mammalian extinctions occurred near the 10 Ka limit of the 15-10 Ka window, which we know was not the case. To reinforce their argument, I think it would be helpful to discuss -- however briefly -- what the expected lag would be for homogenizations following extinction events, and why a millennial-scale lag in these metrics would be expected.

While post-extinction (anthropogenic?) drivers seem like equally plausible candidates for trends in homogenization, it is also not clear to me that the intensification of agriculture is the most likely driver of homogenization during later time bins. Of course extensive agriculture increases during this period, but so do many many other well-documented human-mediated landscape alternations, such as the use of fire. It would be, I think, more circumspect for the authors to consider a larger range of potential drivers rather than just one, the direct evidence for which cannot be evaluated with the data presented here.

If agriculture is truly the driver, would there be obvious regions where agriculture was more widely implemented resulting in increased homogenization vs. regions with lower agricultural activities? If such binning does not result in large differences in homogenization, it would seem that agriculture may not be the central driver.

Specific

L168: what is the metric used to identify 'development of widespread agricultural activities' at 2000-1000 yrsBP? Agricultural activities have been increasing across the world since roughly ca. 7000 yrs BP, so is there a quantitative measure that demarcates these dates relative to other potential markers?

Lines 178-187: it would seem that these documented shifts would benefit from a statistical analysis to relay significant vs. non-significant differences between time periods? The differences in the latter 2 bins are clear, though less so for the 10-5 Ka bin

L263 - Statement seems incorrect: first evidence of increase is in the 10-5 Ka bin... it appears incorrect to extend this to the beginning of the prior bin (15K) where no evidence of increased homogenization is documented.

Fig 1 - difficult to see the 'whiskers' - can these be staggered so that the overlap between different bins is more clear? (Also true for figure 3)

Line 381: the statement that homogenization preceded the modern era by 15000 years seems incorrect based on the findings presented here... the first notable increase occurs within the 10-5 Ka time bin, correct? If Homogenization began at 15 Ka, it would certainly bias the 15-10 Ka time bin

(though if anything homogenization is lower for mammals > 1 Kg during this period).

Reviewer #2 (Remarks to the Author):

The authors incorporated all my previous suggestions. The manuscript can be accepted. Such as I pointed out, this paper is an excellent contribution to the general discussion about the direct and indirect impact of humans on the ecological communities, considering long time intervals and using current and prehistoric data.

Reviewer #3 (Remarks to the Author):

The revised manuscript is greatly strengthened from the original. The Introduction is a much clarified overview of their study, introducing key concepts to the reader and more fully outlining the authors' approach. In general, the study is much more transparent, both in description of methods and making data/code available. The authors have taken on board areas of concern and included further analyses to address these, such as sizes of time bins and issues of measuring climate heterogeneity. They have also more fully considered all of these issues within the Discussion. I am happy to recommend the revised manuscript for publication. Well done to the authors.

Thank you to the reviewers for their time in providing comments on our manuscript. Please find our responses below.

REVIEWER COMMENTS

Reviewer #1 (Remarks to the Author):

Overall, I felt that the additions/edits made to the manuscript clarify the approach, and I commend the authors on their efforts during the revision. I still feel, though, that the link between the observed trends and stated drivers is a bit simplistic and at times over-reaches.

Specifically, the authors' interpretation of their homogenization data, and in particular their emphasis that homogenization began 15 Ka years before present, seems a bit misleading. To me, it is not immediately clear that the Pleistocene extinctions are the foremost culprit of the first noted increase in homogenization within the 10-5 Ka time bin (fig 1). As the authors state, the megafaunal extinctions occur within the 15-11.7 Ka window, yet homogenization does not appear increased during this period. It seems strange to me that there would not be any detected effects of these extinctions during the 15-10 Ka window if extinctions are the primary driver of homogenization... If extinctions are the sole cause, it implies that their effects are lagged enough that systems would not show their effects until thousands of years later... unless all of the mammalian extinctions occurred near the 10 Ka limit of the 15-10 Ka window, which we know was not the case. To reinforce their argument, I think it would be helpful to discuss -- however briefly -- what the expected lag would be for homogenizations following extinction events, and why a millennial-scale lag in these metrics would be expected.

Thank you for this comment. We agree that it is important to clarify in the text.

Firstly, we now make it clearer in the text that there is, in fact, statistical evidence that the majority of megafauna mammal extinctions occur between 12 and 10 ka (see Faith & Surovell, 2009). As such, we are not positing a long delay between the extinction of the mammal megafauna in North America and the appearance of significantly homogenized assemblages.

Secondly, the resolution of the radiocarbon dates from the FAUNMAP database is such that we cannot refine the chronology for the extinctions and biotic homogenization further. What we are confident in is that biotic homogenization (for mammals larger than 1 kg) begins sometime during the 15,000 – 10,000 ybp time bin, resulting in significantly homogenized mammalian assemblages by the 10,000 – 5,000 ybp time bin. Because many of the extinctions occur close to the 10,000 year boundary and we observed significantly homogenized assemblages in the time bin immediately following, we posit that biotic homogenization is linked to the extinction of the mammal megafauna.

While post-extinction (anthropogenic?) drivers seem like equally plausible candidates for trends in homogenization, it is also not clear to me that the intensification of agriculture is the most likely driver of homogenization during later time bins. Of course extensive agriculture increases during this period, but so do many many other well-documented human-mediated landscape alternations, such as the use of fire. It would be, I think, more circumspect for the authors to consider a larger range of potential drivers rather than just one, the direct evidence for which cannot be evaluated with the data presented here.

We now briefly address additional drivers including human population growth, fires, and early agricultural practices. We discuss how human habitation is associated with biotic homogenization through favoring synanthropic species (those who benefit from the results/products of human habitation) and the enhanced fire regimes of the Holocene. We note that fires are very likely not associated with biotic homogenization but rather heterogenization.

If agriculture is truly the driver, would there be obvious regions where agriculture was more widely implemented resulting in increased homogenization vs. regions with lower agricultural activities? If such

binning does not result in large differences in homogenization, it would seem that agriculture may not be the central driver.

Unfortunately, testing this hypothesis is not possible with the data we currently have. Much of the agricultural development between 2,000 and 1,000 ybp occurred in the eastern parts of the USA. We have not performed an additional analysis to address the above comment because sampling of mammal assemblages is not dense enough throughout the entirety of our 30,000 year sample to analyze biotic homogenization for only the eastern states (or even the eastern third of the USA; Fig. S7).

Specific

L168: what is the metric used to identify 'development of widespread agricultural activities' at 2000-1000 yrsBP? Agricultural activities have been increasing across the world since roughly ca. 7000 yrs BP, so is there a quantitative measure that demarcates these dates relative to other potential markers?

The clarification requested here is addressed in two of the cited papers (Marlon et al., 2013; Stephens et al., 2019).

The Stephens et al. (2019) paper uses a crowd sourcing approach wherein the surveyed archaeologists with expertise in assessing land use.

Marlon et al. (2013) use the HYDE dataset (Klein Goldewijk et al., 2010) and the conceptual model of Ruddiman and Ellis (2009).

These sources broadly agree that there is an intensification of land use for agriculture around 1,000 ybp and through to the modern.

We do not believe it is within the scope of the present paper to review the methodologies of previously published papers.

Lines 178-187: it would seem that these documented shifts would benefit from a statistical analysis to relay significant vs. non-significant differences between time periods? The differences in the latter 2 bins are clear, though less so for the 10-5 Ka bin

We have already provided effect sizes in Table 1. They clearly show (bolded) that, for mammals larger than 1 kg, assemblages are more homogenous than expected under our null model by the 10,000 – 5,000 ybp time bin. Similarly, Table 1 shows this is true of all mammals by the 5,000 – 500 ybp time bin.

L263 - Statement seems incorrect: first evidence of increase is in the 10-5 Ka bin... it appears incorrect to extend this to the beginning of the prior bin (15K) where no evidence of increased homogenization is documented.

In the text, we state that biotic homogenization began sometime during the 15 ka – 10 ka time bin, resulting in significantly homogenized faunas by the 10,000 – 5,000 ybp time bin. We believe our analyses support our interpretation. We cannot refine the chronology further due to limitations on the resolution of the radiocarbon dates.

Fig 1 - difficult to see the 'whiskers' - can these be staggered so that the overlap between different bins is more clear? (Also true for figure 3)

We have now staggered the error bars on figure 1.

Line 381: the statement that homogenization preceded the modern era by 15000 years seems incorrect based on the findings presented here... the first notable increase occurs within the 10-5 Ka time bin,

correct? If Homogenization began at 15 Ka, it would certainly bias the 15-10 Ka time bin (though if anything homogenization is lower for mammals > 1 Kg during this period).

What we have intended to state here is that biotic homogenization commenced sometime during the 15,000 – 10,000 ybp time bin, thus producing the significantly homogenized faunas of the 10,000 – 5,000 ybp time bin.

We have changed the specific statement addressed here and in the abstract to say “10,000 years,” however, because we acknowledge there is a difference between a process beginning during a particular time bin and it producing a significant effect by the following time bin.

Reviewer #2 (Remarks to the Author):

The authors incorporated all my previous suggestions. The manuscript can be accepted. Such as I pointed out, this paper is an excellent contribution to the general discussion about the direct and indirect impact of humans on the ecological communities, considering long time intervals and using current and prehistoric data.

Reviewer #3 (Remarks to the Author):

The revised manuscript is greatly strengthened from the original. The Introduction is a much clarified overview of their study, introducing key concepts to the reader and more fully outlining the authors' approach. In general, the study is much more transparent, both in description of methods and making data/code available. The authors have taken on board areas of concern and included further analyses to address these, such as sizes of time bins and issues of measuring climate heterogeneity. They have also more fully considered all of these issues within the Discussion. I am happy to recommend the revised manuscript for publication. Well done to the authors.

We thank Reviewer 2 and 3 for their time in reviewing the newest version of our manuscript.

REVIEWERS' COMMENTS

Reviewer #1 (Remarks to the Author):

As mentioned before, the authors have done a very nice job addressing the concerns raised by myself and the other reviewers... Specifically I think that this revised manuscript is more conservative when attributing different potential causes to the observed patterns, which was my chief concern upon initial receipt. Overall I think it's an interesting paper that adds a lot to our understanding of how communities change in response to large perturbations to the climate and environment. I certainly recommend this revised contribution for publication.